

# Catalog of $NO_x$ emissions from point sources as derived from the divergence of the $NO_2$ flux for TROPOMI

Steffen Beirle[1], Christian Borger[1], Steffen Dörner[1], Henk Eskes[2], Vinod Kumar[1], Adrianus de Laat[2], and Thomas Wagner[1]

[1]Max-Planck-Institut für Chemie (MPI-C), Mainz, Germany
[2]Koninklijk Nederlands Meteorologisch Instituut (KNMI), De Bilt, Netherlands

**Correspondence:** Steffen Beirle
steffen.beirle@mpic.de

**Abstract.**

We present version 1.0 of a global catalog of $NO_x$ emissions from point sources, derived from TROPOMI measurements of tropospheric $NO_2$ for 2018-2019. The identification of sources and quantification of emissions are based on the divergence (spatial derivative) of the mean horizontal flux, which is highly sensitive for point sources like power plant exhaust stacks.

The catalog lists 451 locations which could be clearly identified as $NO_x$ point source by a fully automated algorithm, while ambiguous cases as well as area sources such as Megacities are skipped. 242 of these point sources could be automatically matched to power plants. Other $NO_x$ point sources listed in the catalog are metal smelters, cement plants, or industrial areas. The four largest localized $NO_x$ emitters are all coal combustion plants in South Africa. About 1/4 of all detected point sources are located in the Indian subcontinent and are mostly associated with power plants.

The catalog is incomplete, mainly due to persisting gaps in the TROPOMI $NO_2$ product at some coastlines, inaccurate or complex wind fields in coastal and mountainous regions, and high noise in the divergence maps for high background pollution. The derived emissions are generally too low, lacking a factor of up to 2, mainly due to a low bias of TROPOMI $NO_2$ columns.

Still, the catalog has high potential for checking and improving emission inventories, as it provides accurate and independent up-to-date information on the location of sources of $NO_x$, and thus also $CO_2$.

The catalog of $NO_x$ emissions from point sources is freely available at https://doi.org/10.26050/WDCC/Quant_NOx_ TROPOMI (Beirle et al., 2020).

## 1   Introduction

Nitrogen oxides ($NO_x$=$NO$+$NO_2$) are key species in air pollution and tropospheric chemistry (Seinfeld and Pandis, 2006).

For the prediction of air quality with regional atmospheric chemistry models, accurate and up-to-date $NO_x$ emissions on high spatial resolution are essential (Bouarar et al., 2019). Such data is often difficult to gain for countries with restrictive information





policy. In addition, bottom-up emission inventories take several years to be compiled and are thus generally outdated for countries with quickly developing industrial activities.

Spectrally resolved satellite measurements of solar backscattered radiation allow for the quantification of $NO_2$ and other trace gases absorbing in the UV/vis spectral range by their characteristic spectral absorption structures (Platt and Stutz, 2008;

Richter and Wagner, 2011, and references therein). Tropospheric vertical column densities (TVCDs), i.e. concentrations of $NO_2$ integrated vertically across the troposphere, can be derived by removing the stratospheric contribution and applying the so-called air mass factor (AMF). The AMF can be determined by applying the averaging kernel (AK), reflecting the vertical sensitivity of the satellite measurements, to an a-priori profile of $NO_2$ (Eskes et al., 2003). The AK (and thus the AMF) depends on viewing geometry, surface albedo, aerosols, and particularly on clouds.

$NO_2$ TVCDs from satellite measurements provide independent information on the spatial distribution and strength of tropospheric $NO_2$ levels on global scale since the mid nineties, allowing for the identification of $NO_x$ sources and quantification of $NO_x$ emissions (e.g. Leue et al., 2001; Martin et al., 2003; Mijling and van der A, 2012; Martin, 2008; Monks and Beirle, 2011, and references therein).

In October 2017, the TROPOspheric Monitoring Instrument TROPOMI (Veefkind et al., 2012) was launched as single

payload of ESA's Sentinel-5 Precursor satellite mission. TROPOMI provides $NO_2$ TVCDs on unprecedented high spatial resolution ($7.2 \times 3.6\,\text{km}^2$ until 5 August 2019, $5.6 \times 3.6\,\text{km}^2$ thereafter) and with a high signal to noise ratio (Geffen et al., 2020). Single TROPOMI overpasses clearly reveal $NO_2$ plumes downwind from strong $NO_x$ sources like large power plants (PPs) (Beirle et al., 2019). In temporal mean $NO_2$ TVCDs, however, the high spatial resolution is partly lost due to the averaging over plumes with different directions (related to the variability of atmospheric winds).

Beirle et al. (2019) thus proposed to average $NO_2$ *fluxes* $\boldsymbol{F} = V\boldsymbol{u}$, i.e. TVCDs multiplied with horizontal wind components. Upscaling $NO_2$ to $NO_x$ and applying the continuity equation for steady state, this directly allows for the quantification of $NO_x$ emissions from the divergence, i.e. the spatial derivative of the mean $NO_x$ flux:

$$D := \nabla \cdot \boldsymbol{F} = E - S, \tag{1}$$

with $\boldsymbol{F}$ being the mean $NO_x$ flux, $E$ the $NO_x$ emissions, and $S$ representing $NO_x$ sinks, i.e. the chemical loss of $NO_x$.

In Beirle et al. (2019), maps of $D = \nabla \cdot \boldsymbol{F}$ and $E = D + S$ have been derived and $NO_x$ emissions have been localized and quantified exemplarily for Riyadh, South Africa and Germany. The sink term $S$ was estimated assuming a constant lifetime of $\tau = 4$ hours, as derived from the downwind decay of $NO_2$ for Riyadh (Beirle et al., 2011). For spatially extended sources, like megacities such as Riyadh, $S$ contributes significantly to the derived emissions. For point sources, however, such as the large PPs around Riyadh, emissions are dominated ($\approx 90\%$) by the divergence term $D$ directly: point sources show up as distinct

peaks in the divergence map, which are much sharper than the corresponding peaks in mean TVCD maps, as the $NO_x$ flux increases discontinuously at the $NO_x$ sources, resulting in large values of flux derivatives.

Here we extend this study to global scale, with a particular focus on point sources (PS). Note that due to TROPOMI's spatial resolution of about 5 km, "point sources" could be individual facilities, but also the merged emissions from industrial areas. PS





are identified and quantified based on peaks above local background in divergence maps $D$ directly, rather than emission maps $E$, which allows for clearer identification of PS peaks, as well as for the classification of ambiguous cases by artifacts in $D$.

As the calculation of $D$ from the derivative of mean fluxes requires gridding of TROPOMI data on high spatial resolution, the data processing on global scale is demanding for I/O operations and working memory. Thus, the analysis is only performed
for selected grid pixels around stationary $NO_x$ sources. The selection mask, however, is not prescribed a-priori, but defined based on the magnitude as well as the temporal variability of TROPOMI TVCDs.

From the derived divergence maps, a catalog of $NO_x$ point sources is extracted by a fully automated algorithm.

The manuscript is organized as follows: In section 2, the input data sets used in this study are specified. The detailed data processing is explained in section 3. Section 4 presents the $NO_x$ point source catalog. In sect. 5, the limitations and the potential
of the catalog are discussed, followed by an outlook and conclusions.

## 2   Input data

### 2.1   Tropospheric NO₂ column densities

The $NO_x$ point source catalog is based on $NO_2$ TVCDs from TROPOMI for the years 2018-2019, using the offline product (with successively increasing algorithm version from 0.11.0 on 1 January 2018 to 1.3.0 on 31 December 2019), as provided by
KNMI/ESA via copernicus.eu. Details of the TROPOMI tropospheric $NO_2$ product are given in Geffen et al. (2019), Geffen et al. (2020), and Eskes et al. (2020).

TROPOMI is flying on a sun-synchronous orbit with a local overpass time of about 1:30 p.m. The pixel size at nadir was $7.2 \, km \times 3.6 \, km$ initially, and even improved to $5.6 \, km \times 3.6 \, km$ from 6 August 2019 on (Geffen et al., 2020). TROPOMI provides daily global coverage, resulting in quite good statistics already for annual means.

### 2.2   Meteorological data

Horizontal wind fields $u$ and $v$ as well as air pressure $p$ and air temperature $T$ are taken from reanalysis data from the European Centre for Medium-Range Weather Forecasts (ECMWF). Wind fields are required for the calculation of fluxes. $p$ and $T$ are needed for scaling $NO_2$ up to $NO_x$: $T$ is used for parameterizing the reaction rate constant of $NO+O_3$, and $T$ and $p$ are used to convert $O_3$ mixing ratios from a model climatology to $O_3$ concentrations (see section 3.5).
Until August 2019, ERA-Interim data are used. Since September 2019, ERA-5 data are used (Hoffmann et al., 2019). For both data sets, a preprocessed dataset was created where the 6 hourly model output (0, 6, 12, 18 h UTC) was interpolated to a regular horizontal grid with a resolution of $1°$. For future processing, sampling will be based on finer temporal and spatial grids.





**Table 1.** Processing settings in this study as compared to Beirle et al. (2019).

| Section | Procedure | This study | Beirle et al. (2019) |
|---|---|---|---|
| 3.1 | $NO_2$ selection criteria | qa > 0.75, CF<0.3 | qa > 0.75, CF<0.3 |
| | | 2018-2019 | Dec 2017 - Oct 2018 |
| | | SZA < 65° | April to October (Germany) |
| 3.2 | Grid | 0.025° | 0.027° |
| 3.3 | Regions of interest | 61° S to 61° N, | Riyadh, South Africa, Germany |
| | | masked for stationary sources | |
| 3.4 | Interpolated wind fields | 300 m above ground | fixed vertical level at about 450 m |
| 3.5 | $[NOx]/[NO2]$ | photo-stationary state | $1.32 \pm 20\%$ |
| 3.6 | Background correction | none | 5th percentile |
| 3.6 | Lifetime correction | none | $\tau$=4 h |
| 3.7 | AMF correction | none | up to factor 2 for Germany |
| 3.9 | Peak fit | | |
| | Iteration | automated | semi-automated |
| | Pre-classification | multi step | none |
| | Fit function | linear background + 2-D Gaussian | linear background + 2-D Gaussian + rotation |
| | Peak removal | fitted peak $\pm 2\sigma$ set to NaN | fitted peak subtracted |

## 2.3 Ozone climatology

Ozone mixing ratios, used for the scaling of $NO_2$ to $NO_x$, were taken from the Earth System Chemistry integrated Modelling (ESCiMo) project (Jöckel et al., 2016), using the RC1SD-base-10a simulation for the years 2000-2010. The monthly mean climatology was calculated from the model fields sampled online along the OMI-Aura overpass time (which is close to the TROPOMI overpass time) using the MESSy SORBIT submodel (Jöckel et al., 2010).

## 2.4 Power plant database

The Worlds Resources Institute provides an open access Global Power Plant Database (GPPD) (Byers et al., 2019). We use this database in order to automatically identify $NO_x$ point sources corresponding to PPs.

The GPPD lists almost 30,000 power plants of all kinds, including solar-, nuclear-, and hydro power. For our purpose, we created a subset of those PPs using coal, gas, or oil as primary fuel. In addition, PPs with capacities below 100 MW are skipped. The resulting subset of GPPD comprises 4654 PPs of which 2013, 2265, and 376 use coal, gas, and oil as primary fuel, respectively.




## 3 Data processing

In this section we describe the data processing step by step. Table 1 summarizes the main steps and also lists similarities and differences to the procedure described in Beirle et al. (2019).

### 3.1 NO$_2$ selection

For this study, we select TROPOMI tropospheric NO$_2$ column densities $V_{NO_2}$ for the years 2018-2019 with values of the data quality indicator ("qa value") above 0.75, as recommended in Geffen et al. (2019), and effective cloud fractions (CF) below 0.3. These selection criteria are the same as in Beirle et al. (2019).

In addition, we skip measurements with SZA above 65° for the calculation of fluxes. This strict criterion removes observations for sun being low, and implicitly results in a gradual removal of wintertime measurements for mid-latitudes, while in
Beirle et al. (2019), winter months have been skipped explicitly for Germany. Wintertime measurements are skipped in order to avoid unfavorable viewing conditions, snow covered scenes, and stronger interference with aged plumes due to longer lifetimes. Moreover, the SZA restriction allows to simply parameterize the NO$_2$ photolysis as function of the SZA (see section 3.5).

### 3.2 Grid

For each TROPOMI orbit, $V_{NO_2}$ are gridded to a regular longitude/latitude grid with 0.025° resolution for 61° S to 61° N. Note that there are a few small NO$_x$ sources North of 61°, but due to the strict SZA threshold of 65°, the flux statistics would be poor for higher latitudes.

Gridding is based on linear interpolation of TROPOMI pixel centers using the griddata function from the Python module SciPy (Virtanen et al., 2020). This approach allows for fast gridding. In addition, there are no discontinuities at the TROPOMI
pixel borders, which would lead to extremely high (positive and negative) values of the derivative.

All missing values (qa<0.75) as well as the outermost pixels on each side of the TROPOMI swath (i.e. the pixels with the highest viewing zenith angles) are set to not a number (NaN). This is necessary in order to restrict the area of interpolated TVCDs to the actual area covered by measurements.

Hereafter we denote the longitude and latitude dimensions as $x$ and $y$, respectively, in vector indices as well as in the text.

### 3.3 Considered pixels

The detection of NO$_x$ point sources from the divergence method requires a high spatial resolution of some km. Having such a fine grid on (almost) global scale is quite demanding for I/O operations and working memory. But large parts of the globe are free from stationary NO$_x$ point sources, in particular oceans, deserts and forests. In order to reduce processing time, we thus define a mask for pixels to be processed. However, this mask is not defined a-priori, but based on a pre-selection of potential
stationary sources as derived from TROPOMI data.

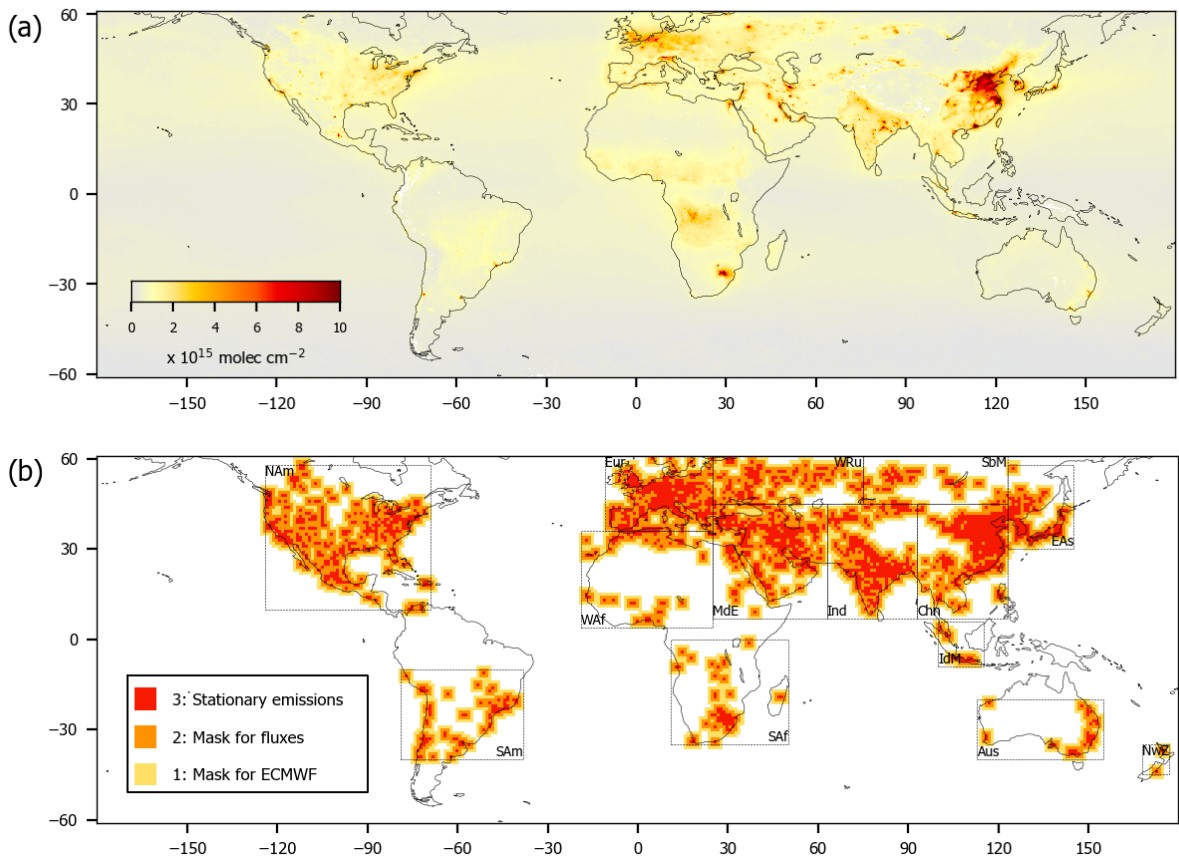

**Figure 1.** (a) Mean tropospheric NO₂ column density for 2019. (b) Derived mask **M** for the selection of pixels investigated in this study. Boxes indicate the regions as defined in table 2.

In a first step, we calculate maps of monthly mean, minimum and maximum TVCDs for 2018-2019 (including SZA up to 80°). Fig. 1(A) displays the annual mean for 2019. Note that due to long range transport or potential bias in the stratospheric correction, TVCDs might be enhanced for regions without local emissions. In addition, regions exposed to seasonal biomass burning or lightning can show enhanced NO₂ columns that are not caused by stationary point sources. Thus, the mask is not
5  just defined based on an absolute column threshold.

Instead, we define pixels that likely contain stationary sources based on the following three criteria, where thresholds have been derived empirically such that industrial regions are kept while non-stationary sources like biomass burning regions are removed:

1. Sufficient statistic: Monthly means (based on at least 5 valid TROPOMI overpasses) must be available for at least 6
10      months during 2018-2019.



**Table 2.** Definition of regions used for the regional statistics shown in table 3 and for regional figures shown in the Supplement.

| Label | Region[1] | lon [°E] | lat [°N] |
|---|---|---|---|
| NAm | North America | -124 to -69 | 10 to 58 |
| SAm | South America | -79 to -38 | -40 to -10 |
| Eur | Europe | -11 to 25 | 36 to 61 |
| WAf | West Africa | -19 to 25 | 4 to 36 |
| SAf | South Africa | 11 to 50 | -35 to 0 |
| WRu | West Russia/East Europe | 25 to 75 | 45 to 61 |
| SbM | Siberia/Mongolia | 75 to 123 | 45 to 61 |
| MdE | Middle East | 25 to 63 | 7 to 45 |
| Ind | Indian subcontinent/West China | 63 to 93 | 7 to 45 |
| Chn | East China/South East Asia | 93 to 123 | 7 to 45 |
| EAs | East Asia | 123 to 145 | 30 to 58 |
| IdM | Indonesia/Malaysia | 100 to 115 | -9 to 6 |
| Aus | Australia | 113 to 155 | -40 to -20 |
| NwZ | New Zealand | 168 to 177 | -45 to -35 |

[1] Note that the regions are defined such that all considered pixels are covered by a limited number of figures with similar area as far as feasible. Region names are mostly based on continents. For Asia, regions are labeled after the countries dominating the detected point sources, gaining tangibility while condoning some inaccuracies in actual country borders.

2. Local maximum: the mean TVCD for 2019 must be enhanced by more than $0.5 \times 10^{15}$ molec cm$^{-2}$ compared to the local background. This criterion is sensitive for all kind of NO$_x$ sources, including biomass burning.

3. High dynamical range all over the year: The difference between monthly maximum and minimum must exceed $1.5 \times 10^{15}$ molec cm$^{-2}$ for at least 90% of the available months.

   Due to changing wind patterns, a high dynamical range of TVCDs is expected next to stationary sources. For biomass burning, this is only fulfilled during burning season. This criterion thus removes biomass burning regions.

All pixels fulfilling these criteria are considered as candidates for stationary NO$_x$ sources. We now define a mask **M** on 1° resolution, where every grid pixel $i, j$ containing a stationary source candidate is set to $M_{i,j}=3$.

As the further analysis is based on peaks in the divergence map around point sources, also the surrounding of stationary sources should be kept. Thus, the mask values for all neighboring 1° pixels $i \pm 1, j \pm 1$ with $M_{i,j}=3$ are set to 2 (if not set to 3 already), and all pixels with $M_{i,j} \geq 2$ are considered for the further analysis of fluxes and divergence. Note that even at 60°N, this corresponds to adding more than 50 km in longitude, which is far more than the 30 km radius considered for peak fitting below (sect. 3.9.2).





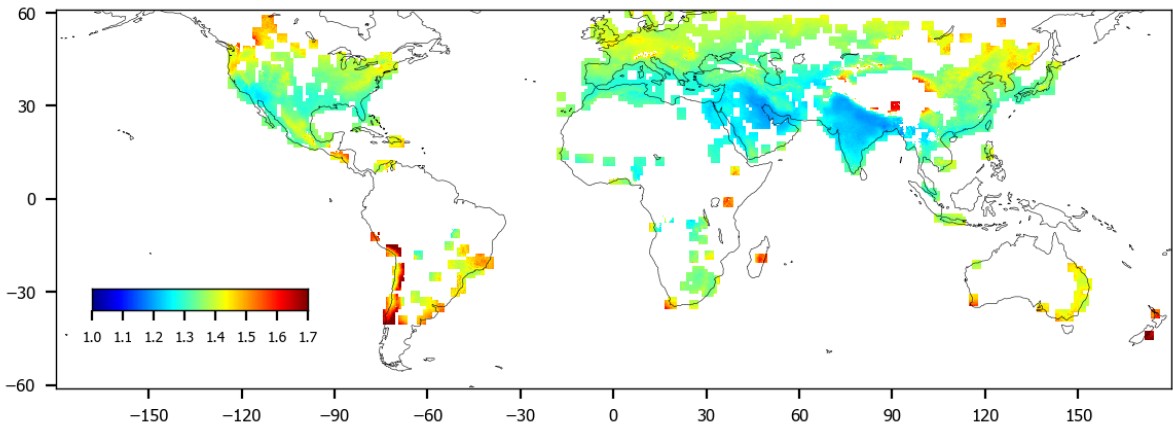

**Figure 2.** Ratio of mean tropospheric $NO_x$ (as derived by assuming photostationary state according to Eq. 2 for each TROPOMI pixel) and mean $NO_2$ column density over pixels with **M**$\geq$ 2. for 2018-2019. High values over mountains are caused by lower values for both $k$ (due to colder Temperatures) and $[O_3]$ (due to lower pressure).

For all second next neighboring pixels $i \pm 2, j \pm 2$, mask values are set to 1 (if not set to 2 or 3 already). Values $M_{i,j} \geq 1$ are used for the extraction of ECMWF data, which are required for a larger area to enable spatial interpolation later on.

The resulting mask **M** is displayed in Fig. 1(B). In addition, it is provided as datafile in the Supplement. Most parts of the continents are kept, except deserts and forests. Open oceans are completely skipped. The application of the mask reduces the amount of pixels included in the further analysis to 13.5% and thereby enables the processing of fluxes on a common desktop computer.

### 3.4 Meteorological data

The meteorological datasets $u$, $v$, $p$, and $T$ are extracted from ECMWF input data by linear interpolation in three steps:

1. in vertical dimension to an altitude of 300 m above ground for each ECMWF input dataset with 6 hourly resolution. As the focus of this study are emissions from point sources, we consider wind fields representative for the transport of freshly released $NO_x$ emissions from stacks. The choice of altitude of wind fields is further discussed in sect. 5.2.2.

   Only pixels with the mask being 1 or higher are extracted.

2. in time dimension to the orbit time stamp of each TROPOMI orbit, as given in the orbit filename. The actual TROPOMI overpass lags the orbit time stamp by about 33 and 67 minutes at 60° S and 60° N at solstice $\pm$ 7 minutes for summer/winter. Thus the orbit time stamp reflects the wind conditions for recent plume histories.

3. in lat/lon to the 0.025° grid.





## 3.5 Up-scaling of $NO_2$ to $NO_x$

In this study we scale the measured $NO_2$ TVCD to a $NO_x$ TVCD for each TROPOMI pixel. The conversion factor $L$ is calculated according to the photostationary steady-state

$$L := \frac{[NO_x]}{[NO_2]} = 1 + \frac{[NO]}{[NO_2]} = 1 + \frac{J}{k[O_3]} \qquad (2)$$

The impact of VOCs is neglected here as the focus is put on $NO_x$ point sources and thus generally high $NO_x$ concentrations. The photolysis frequency of $NO_2$ $J$ is parameterized as function of the SZA $\theta$ by

$$J = 0.0167 \, \exp(-0.575/\cos\theta) \, s^{-1} \qquad (3)$$

as proposed by Dickerson et al. (1982). This parameterization is "accurate to about 10% for mostly sunny conditions" for SZA $< 65°$ (Dickerson et al., 1982).

The reaction rate constant $k$ for the reaction of NO with $O_3$ is parameterized as function of temperature (in Kelvin) by

$$k = 2.07 \times 10^{-12} \times \exp(-1400/T), \qquad (4)$$

following the recommendations from Atkinson et al. (2004) and IUPAC (2013).

Ozone mixing ratios are taken from a climatology based on the ESCiMo model simulation (2.3) and converted into concentrations based on $T$ and $p$ from ECMWF.

The derived values for $L$ represent conditions for surface-near pollution. For background $NO_x$ in the upper troposphere, the partitioning would be shifted towards NO. However, any additive background is automatically removed by the calculation of the divergence.

Fig. 2 displays the ratio of temporal means of $NO_x$ and $NO_2$ TVCDs. Global mean is 1.35 with a SD of 0.08. Values for Riyadh, South Africa and Germany are 1.22, 1.36 and 1.41, respectively, in agreement with the value of 1.32±0.26 applied
in Beirle et al. (2019), which was based on the number given in Seinfeld and Pandis (2006) for polluted conditions around noontime.

Note that the actual $[NO_x]/[NO_2]$ ratio close to a point source might be different in case of high NO concentrations causing $O_3$ titration. In this case, however, the divergence method would detect the emitted $NO_x$ downwind from the source as soon as the NO is converted to $NO_2$ after mixing with ambient air. This results in a spatial smearing of the peak in the divergence map,
leading to broader peaks, but the same integral (and thus emissions) for the peak fitting algorithm (section 3.9.2). For the final budget of $NO_x$ emissions, which are determined from the integrated peaks, the final photostationary state is thus still adequate.

## 3.6 Lifetime and background corrections

In Beirle et al. (2019), emission maps were derived by adding the sink term $S = V/\tau$ to the divergence map. For this, a constant lifetime of $\tau = 4\,h$ was applied, as derived from OMI data for Riyadh (Beirle et al., 2011). In addition, $V$ was corrected by
subtracting the regional background, as the lifetime estimate was derived for freshly released, surface near pollution, while upper tropospheric background $NO_2$ generally has a longer lifetime.





As discussed in Beirle et al. (2019), the inclusion of the sink term has significant impact on area sources; it contributes about 50% of integrated emissions for Riyadh urban area. For point sources, however, the emission signal is by far dominated by the divergence term, for instance accounting for 87% of the emissions of PP9 in Beirle et al. (2019).

Within this study, we do not correct for the sink term $S$ for the following reasons:

– The $NO_x$ lifetime is expected to be different for the diverse conditions in the considered regions, covering a large variability of temperature, humidity, actinic flux, volantile organic compounds (VOC) levels, and $NO_x$ levels.

– $NO_x$ lifetimes simulated from global models cannot resolve the nonlinearities caused by point sources. In addition, emission inventories used as input to model runs generally have a time lag, thus emissions are outdated for quickly developing countries. Consequently, modeling the $NO_x$ lifetime for the investigated point sources is challenging and
uncertain.

– Identifying point sources in the divergence map $D$ directly is more immediate than in $E = D + S$, as point sources reveal sharper peaks in $D$ than in $E$ (Fig. 2 in Beirle et al., 2019). In addition, the identification of ambiguous candidates as e.g. caused by inaccurate wind fields is clearer based on $D$ directly (sect. 3.9.1).

The resulting low bias of point source emissions caused by the missing lifetime correction is discussed in sect. 5.2.1.
Since no lifetime correction with $S = V/\tau$ is performed, also the background correction for $V$, which was performed in Beirle et al. (2019) in order to exclude upper tropospheric $NO_x$ with longer lifetime and different $L$, is omitted in the current study. Note that the local background of $V$, as well as any potential offset due to e.g. stratospheric correction, would affect the lifetime correction, but have no impact on $D$, as any additive term is lost by the calculation of the derivative.

## 3.7 AMF correction

Tropospheric column densities of $NO_2$ are derived from the total slant column by subtracting the stratospheric column and applying the so-called air mass factor (AMF), representing the sensitivity of the satellite measurements for tropospheric $NO_2$ (Eskes et al., 2003). The AMF mainly depends on the a-priori vertical profile and cloud properties.

Validation studies report on a general low bias of $NO_2$ TVCD from TROPOMI (e.g. Verhoelst et al., 2020; Judd et al., 2020), caused by a high biased AMF. Part of this bias seems to be related to the a-priori profiles which do not resolve the
pollution profiles close to sources. In addition, there are indications that the cloud heights used for the TVCD retrieval are biased low and the albedo maps used are biased high (Eskes et al., 2020).

In Beirle et al. (2019), an AMF correction was performed for South Africa and Germany by applying the provided AK to a surface-near profile. In this study, we do not apply such an AMF correction, as the effects of biased input albedo and cloud height cannot be corrected a-posteriori based on the AKs, but require a reprocessing of the TROPOMI $NO_2$ product.
Consequently, the low bias of TROPOMI TVCDs is directly transferred into the $NO_x$ emissions listed in the catalog, as discussed in detail in sect. 5.2. The low bias is expected to be improved with an updated $NO_2$ product, which will then be used for deriving an updated version of the point source catalog.



## 3.8 Gridded fluxes and divergence

From gridded $NO_x$ columns and gridded wind fields, the gridded $NO_x$ flux in both $x$ and $y$ direction is derived for each TROPOMI orbit. Mean fluxes are calculated for the period 2018-2019. For grid pixels with less than 25 measurements, fluxes are set to missing values due to poor statistics. Note that we do not explicitly skip winter months for midlatitudes, as in Beirle et al. (2019) for Germany, but they are removed implicitely by the strict SZA threshold of $65°$.

From the mean zonal and meridional flux maps, the divergence map $D = \nabla \cdot \boldsymbol{F}$ is calculated, which is the basis for the identification and quantification of point sources below.

## 3.9 Iterative peak fitting

We apply a fully automated iterative peak fitting algorithm in order to detect $NO_x$ point sources. The goal is to identify clear point source peaks in the divergence map, where a robust quantification of emissions is possible, while ambiguous cases are skipped. Thus, the resulting catalog of point sources is incomplete; a detailed discussion on various reasons for missing PS is given in section 5.1. But the remaining PS listed in the catalog correspond to actual $NO_x$ sources with high confidence.

In each iteration, the following procedure is executed:

1. The grid pixel with highest value of the divergence is considered as point source *candidate*.

2. Each candidate is classified into different categories, and skipped if ambiguous (sect. 3.9.1).

3. For a promising candidate, a 2-D Gaussian is fitted to the divergence peak, and successful fits are included in the catalog (sect. 3.9.2).

4. The candidate is removed from the divergence map before searching for the next highest $D$ value (sect. 3.9.3).

The iteration stops as soon as the maximum value of the divergence is less than 0.2 $\mu g \, m^{-2} \, s^{-1}$, which is $< 4\%$ of the global maximum value. Below this threshold, almost no further point sources could be detected which meet the quality criteria listed below. For future versions, the availability of several years of TROPOMI data is expected to decrease the noise in divergence maps and will probably allow to decrease this threshold and investigate additional small $NO_x$ point sources.

### 3.9.1 Pre-classification of candidates

Point source candidates are iteratively defined as the location of maximum divergence in the global map. Before fitting a Gaussian peak, and quantifying emissions, however, artifacts and ambiguous cases have to be excluded.

For this, a pre-classification is done based on zooms of the divergence map 30 km around the candidate. As soon as the candidate is classified in one of the following categories, the pre-classification stops. In Fig. 3, examples for each category are shown.

1. Category *gap*:
   If more than 25% of grid pixels are missing within 8 km around the candidate, it is classified as "gap". Gaps were found





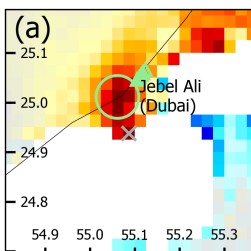 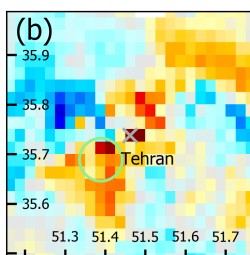 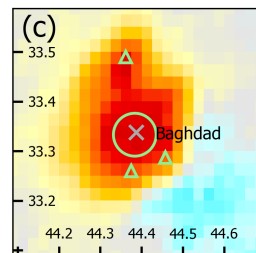 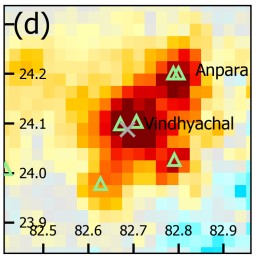 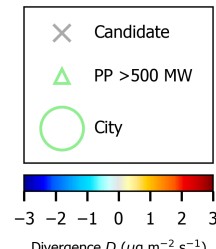

**Figure 3.** Zooms of the divergence map showing examples for the different categories of point source candidates checked during pre-classification. White indicates missing data. The location of PPs from GPPD as well as cities is indicated by green markers. (a) "Gap" candidate near Jebel Ali/Dubai. (b) "Negative" candidate near Tehran. (c) "Area source" around Baghdad. (D) "Area source" candidate covering several power plants (with Vindhyachal and Anpara being the largest) in India.

primarily at sandy coastlines and are caused by persistent cloud coverage above threshold, which is probably an artifact of the coarse resolution of the albedo map used for the cloud retrieval. In the later stage of the iteration, gaps also occur in the vicinity of strong point sources which have been already removed from the divergence map. Figure 3(A) displays an example for a candidate of category "gap" around Dubai, where the global maximum of divergence was found, but missing data does not allow for further quantifications.

2. Category *negative*:

If the minimum divergence within the 30 km zoom is negative with an absolute value larger than 50% of the maximum value, the candidate is "negative".

Negative values of the divergence generally indicate $NO_x$ sinks. Thus values of $D < 0$ have to be expected downwind from sources. But absolute values should be far lower than the positive values at the place of emissions, as the $NO_x$ removal is taking place over large distances (at wind speed of 5 m/s, a lifetime of 4 hours corresponds to an e-folding distance of 72 km).

High absolute values of negative divergence thus cannot be explained by chemical loss of $NO_x$, but indicate an inappropriate simplification of the complex 3D wind fields by a 2D wind vector on rather coarse spatial resolution. Also changes of $NO_x$ emissions or wind patterns on temporal scales of some hours (whereas Eq. 1 assumes steady state, i.e. neglects of the temporal derivative of $V$) can cause high negative values of $D$.

Figure 3(B) displays an example of high negative divergence around Tehran. Obviously, the divergence method fails here, even though TVCDs show a hotspot of very high $NO_2$ around Tehran that can even be spotted in the global map (Fig. 1(b)). Reason for the noisy divergence is the location of Tehran next to the Alborz mountains, where actual wind patterns are not described appropriately by the low-resolution wind fields.





3. Category *area source*:

   If the candidate is neither classified as "gap" nor as "negative", sections of zonal and meridional means are calculated in order to allow for a quick check of the spatial extent of the divergence peak. For both sections, the full width half maximum (FWHM) $W_{\text{first guess}}$ is determined by checking for the first occurrence of a value below half of the maximum in both directions. For point sources, a FWHM of about 12 km has been reported in Beirle et al. (2019). Values of $W_{\text{first guess}}$ above 17 km in both $x$ and $y$ thus indicate a $NO_x$ source which cannot be categorized as single point source, but an area source, which could be a large city or an extended industrial area with several point sources nearby. Figure 3(C) and (D) display two examples for candidates classified as area source. (C) shows the city of Baghdad. In (D), an "area source" consisting of several point sources close to each other is shown, primarily the coal fired power plants Vindhyachal (5 MW) and Anpara (2 MW) in India.

### 3.9.2 Gaussian fit

If a candidate passes all pre-classification checks, a 2-D Gaussian on top of a linear background is fitted to the peak in the divergence map:

$$G(x,y) = \frac{A}{2\pi\sigma_x\sigma_y} \exp\left(\frac{-(x-\Delta x)^2}{2\sigma_x^2}\right) \exp\left(\frac{-(y-\Delta y)^2}{2\sigma_y^2}\right) + m_x x + m_y y + b, \tag{5}$$

with the fit parameters $A$ being the peak integral, $\sigma_x, \sigma_y$ the width of the Gaussian, $\Delta x, \Delta y$ the shift of the peak maximum (relative to the first guess candidate location corresponding to the highest $D$ value), and $m_x, m_y$ and $b$ describing a linear background.

In contrast to Beirle et al. (2019), no rotation of the peak is allowed in order to make the fit fast and stable, and in order to be able to interpret the widths $\sigma_{x,y}$ as actual distance in latitudinal and longitudinal dimensions.

The parameters are determined by a least-squares fit of $G$ to the divergence map within 22 km around the candidate. As starting values, $\sigma_{x,y}$ are set to $W_{\text{first guess}}/2.355$ for both $x$ and $y$. But since the fit yields a more robust measure of the peak width than the simple FWHM estimate determined during pre-classification, the candidate is again classified as area source, if $W_{\text{fit}} := 2.355 \times \sigma_{x,y}$ exceeds 17 km for $x$ or $y$.

Otherwise, the candidate is considered to be a point source, where the fitted parameter $A$ of the Gaussian peak represents the corresponding PS emissions. However, in order to only keep robust emission estimates in the point source catalog, cases with emissions below 0.03 kg/s (which has been derived in Beirle et al. (2019) as detection limit for optimal conditions) or a relative fit error above >30% are categorized as "uncertain".

### 3.9.3 Candidate removal

The candidate has to be removed from the global map before the next iteration step. Removal is implemented by setting the divergence values around the maximum to NaN. Note that in Beirle et al. (2019) the fitted peaks were subtracted instead. However, this would introduce a highly structured residue in the divergence map, which would create several new artificial





candidates for the fully automated peak search algorithm. Removing candidates by setting $D$ to NaN avoids such artificial point sources and prevents any later interferences from fit residues from neighboring sources.

Depending on the classification, the following procedure is applied:

– For point sources, an ellipse with $2 \times \sigma_{x,y}$ as semi major/minor axis is removed, with $\sigma$ from the Gaussian fit.

5 – For categories "gaps" and "uncertain", all pixels within 22 km around the maximum are removed.

– For category "negative", a larger area (30 km around the maximum) is removed, as negative artifacts generally occur not at, but next to the sources.

– For area sources, also 30 km around the maximum are removed, if classified based on $W_{\text{first guess}}$. If the area source was classified based on $W_{\text{fit}}$, an ellipse with $2 \times \sigma_{x,y}$ is removed as for point sources.

## 10 3.10 Identification of point sources associated with power plants

We perform an automated match of the point source catalog with the combustion PPs listed in GPPD. For each point source, we search for GPPD entries within 5 km radius. In the point source catalog, we add

– the integrated capacity of all PPs within 5 km,

– a complete list of the names of all PPs within 5 km, and

15 – the primary fuel of the largest PP within 5 km.

## 4 Results

### 4.1 Candidate classification

The iterative peak fitting algorithm yields 7250 candidates, of which 451 are classified as point sources. For 242 of these point sources, a match with GPPD power plants was found. Table 3 lists the classification statistics for the regions defined in table 2.

20 Figure 4 displays regional maps of color-coded $D$ for zooms of selected regions. The respective maps for all regions listed in table 2 are provided in the supplement in pdf format allowing for loss free zooming.

The candidate classifications are indicated by symbols, where point sources with/without a PP match are displayed as triangles in magenta/dark grey, respectively. Non-point sources are shown in light grey. For sake of clarity, they are only displayed for high divergence values ($D_{\max} > 1 \mu\text{g m}^{-2}\text{ s}^{-1}$ or integrated $D > 0.1 \text{ kg s}^{-1}$), as they would otherwise dominate 25 the figure for some regions like Europe, where 499 candidates are classified as "gap" and 1013 as "negative" (table 3).

The map of the Middle East (A) contains most of the non-point source examples shown in Fig. 3. Several more gaps occur all along the Persian Gulf coastline, and further negative candidates are found in mountains as well as along the coastlines.

Several large cities like (A) Cairo and Jeddah, (B) Paris and Madrid, (C) Delhi and Mumbai, or (D) Moscow and Saint Petersburg are categorized as area source. However, there are also some candidates categorized as area source which do not

**Table 3.** Number of candidates and their respective classification found for the regions defined in table 2. The number of PS in brackets refers to the point sources associated with power plants.

| Label | Region | Candidates | thereof: PS (PP) | "gap" | "neg" | "area" | "uncertain" |
|---|---|---|---|---|---|---|---|
| NAm | North America | 880 | 47 (32) | 171 | 601 | 41 | 20 |
| SAm | South America | 172 | 8 (2) | 32 | 117 | 11 | 4 |
| Eur | Europe | 1558 | 24 (11) | 499 | 1013 | 20 | 2 |
| WAf | West Africa | 116 | 19 (6) | 30 | 54 | 10 | 3 |
| SAf | South Africa | 171 | 16 (9) | 43 | 97 | 9 | 6 |
| WRu | West Russia/East Europe | 469 | 41 (23) | 84 | 319 | 19 | 6 |
| SbM | Siberia/Mongolia | 186 | 9 (6) | 45 | 123 | 5 | 4 |
| MdE | Middle East | 721 | 107 (40) | 245 | 305 | 43 | 21 |
| Ind | India/Pakistan/West China | 493 | 114 (76) | 129 | 198 | 36 | 16 |
| Chn | East China/South East Asia | 1766 | 34 (16) | 678 | 1013 | 36 | 5 |
| EAs | East Asia | 561 | 19 (10) | 139 | 378 | 20 | 5 |
| IdM | Indonesia/Malaysia | 65 | 5 (4) | 25 | 34 | 1 | 0 |
| Aus | Australia | 66 | 7 (7) | 16 | 34 | 7 | 2 |
| NwZ | New Zealand | 11 | 0 (0) | 1 | 10 | 0 | 0 |
| Glb | Global | 7250 | 451 (242) | 2139 | 4308 | 258 | 94 |

correspond to a megacity. In particular the candidate corresponding to the maximum divergence over India, which is caused by the coal-fired 5 GW Vindhyachal Super Thermal Power Station, was categorized as area source, as it interferes with the 4 GW Anpara power plant about 16 km Northeast (Fig. 3 D). Such sources could still be investigated in detail based on the divergence map. However, for interfering sources such close to each other, a quantification by a fully automated algorithm is challenging.

5  For Riyadh, the power plants 9 and 10 northeast and southeast of the city center are identified as point sources (compare Beirle et al. (2019)). In contrast to Beirle et al. (2019), PP8 west of Riyadh is not identified as PS as it could not be separated from Riyadh city, as a consequence of the strict pre-selection of candidates and the slightly larger fit interval, which were necessary in order to run the algorithm fully automated globally.

Several point sources are detected in the Middle East. There is a remarkable cluster of several point sources detected south of 
10  Baghdad. Note that there was even a point source detected within the Persian Gulf, which corresponds to the Zakum offshore oilfield.

The Indian subcontinent reveals the highest number of point sources of the investigated regions, contributing 1/4th of the global number. This reflects the quickly growing industrial activities, while measures for emission reduction still need to catch up. In addition, the divergence method obviously works very well for India.



**Figure 4.** Divergence $D$ (color coded) and point source classification (symbols) for the (A) the Middle East, (B) India, (C) Europe, and (D) Ukraina/Western Russia. Triangles display the point sources listed in the catalog, where matches to GPPD power plants are indicated in magenta. Regions are zoomed for clarity. Respective maps for all regions defined in table 2 are provided in the Supplement (Figures S1-S14). Also for sake of clarity, non-pointsources are only shown for candidates with $D_{\max} > 1 \mu \mathrm{g}\, \mathrm{m}^{-2}\, \mathrm{s}^{-1}$ or the integrated divergence within 30 km exceeding 0.1 kg s$^{-1}$.





In Europe, only very few PS are detected, like the world's largest charcoal power plant Belchatow in Poland, the German charcoal power plants Jänschwalde, Boxberg, and Neurath/Niederaußem (compare Beirle et al., 2019), or Europe's largest steel plant in Taranto/South Italy. Remarkably, almost no point sources are detected for England and the Benelux countries, where the mean TVCD has a local maximum (Fig. 1 A). Instead, there are several candidates classified as gaps and negative, which
is related to the high noise observed in the divergence map. A similar situation is found for China, where TVCD is highest globally (Fig. 1 A), but noise in $D$ is large (Fig. S10 in the Supplement) such that the number of negative candidates is high (1013), but only few (34) point sources could be clearly identified. In Ukraina and West Russia, where mean TVCD levels are moderate, several point sources could be clearly identified. These striking regional differences in the performance of the automated point source detection will be discussed in detail in sect. 5.1.

**4.2    Catalog of point sources**

We derive a global catalog of $NO_x$ emissions from the 451 peaks categorized as point sources by sorting them according to the fitted emissions. A complete list of all detected point sources, including lat/lon and the estimated emissions, is provided in csv format at https://doi.org/10.26050/WDCC/Quant_NOx_TROPOMI (Beirle et al., 2020).

Table 4 lists a selection of the identified point sources with some additional information on the respective $NO_x$ source. It
contains the top ten emitters of the catalog. In addition, every 100th rank is included in order to illustrate conditions for lower divergence levels. Divergence maps for the same selection are shown in Fig. 5, where also external information on $NO_x$ sources have been added.

Most of the point sources listed in table 4 can actually be associated to single or groups of PPs. Overall, a PP match was found for 242 point sources. The median distance between GPPD and PS locations was found to be 1.6 km, which is better
than TROPOMI resolution. For the selection in table 4, we did some additional inquiry and could identify the PPs Medupi (#4) and Presidente Vargas (#300), both missing in GPPD, as probable $NO_x$ source.

Other point sources are the Secunda CTL coal liquifying facility (#2), steel work facilities (#6, #10), and a cement plant (#100). PS #7 is located in Ulsan, an industrial hotspot in South Korea. Here, however, we could not identify a single dominating $NO_x$ point source. For the Hermosillo PP, the peak fit is probably affected by Hermosillo city nearby (0.7 M inhabitants).

The four highest point source $NO_x$ emissions are all found for South African coal PPs, which have already been presented in Beirle et al. (2019). Note that the emissions in table 4 are lower than those given in Beirle et al. (2019) for various reasons, as discussed in detail in section 5.2.5, mainly due to the missing AMF and lifetime corrections in the current study.

The thresholds for artifacts in divergence have been defined rather strictly. Consequently, the remaining locations listed in the catalog actually indicate stationary $NO_x$ emissions. In the spot tests investigated exemplarily, we found no indication for
false signals in the catalog.

**4.3    Comparison to Power Plant database**

Fig. 6 (a) provides a scatter plot of PP capacity and PS emissions. Respective regional figures for all considered regions are provided in Fig. S15 in the Supplement.





**Table 4.** Extract of the point source catalog for the top ten and every 100th rank. Capacity, fuel type and facility names are added for matches to GPPD. The last two columns are not part of the catalog, but have been added manually for the presented selection in order to provide information on the likely $NO_x$ source where no (or insignificant) GPPD match has been found. Zooms of the divergence map for the same selection are displayed in Fig. 5. As discussed in detail in section 5.2.5, the given emissions are biased low.

| from point source catalog: | | | | | | | additional information: | |
| Rank | Lat | Lon | Emissions | PP capacity | PP fuel[1] | PP name(s)[2] | Country | Other sources |
| | [° N] | [° E] | [kg/s] | [GW] | | | | |
| 1 | -26.284 | 29.176 | 0.886 | 6.600 | Coal | Matla; Kriel | South Africa | |
| 2 | -26.566 | 29.181 | 0.679 | | | | South Africa | Secunda CTL coal liquifyer |
| 3 | -23.686 | 27.594 | 0.669 | 3.990 | Coal | Matimba | South Africa | |
| 4 | -27.104 | 29.788 | 0.668 | 4.110 | Coal | Majuba | South Africa | Medupi PP[3] |
| 5 | 22.397 | 82.692 | 0.588 | 4.830 | Coal | Korba | India | |
| 6 | 40.637 | 109.739 | 0.528 | 0.200 | Coal | Baotou | China | Steel works |
| 7 | 35.502 | 129.303 | 0.523 | | | | South Korea | Ulsan industrial area |
| 8 | -26.777 | 29.379 | 0.474 | 3.654 | Coal | Tutuka | South Africa | |
| 9 | 28.696 | 48.334 | 0.460 | 6.905 | Gas | Az Zour | Kuwait | |
| 10 | 34.930 | 127.723 | 0.460 | 1.330 | Gas | Gwangyang | South Korea | Steel works |
| 100 | 29.009 | 31.216 | 0.151 | | | | Egypt | Cement plant |
| 200 | 44.669 | 89.089 | 0.093 | 7.000 | Coal | Wucaiwan | China | |
| 300 | -22.537 | -44.121 | 0.062 | | | | Brasil | Presidente Vargas PP[3] |
| 400 | 29.099 | -110.988 | 0.040 | 0.250 | Gas | Hermosillo | Mexico | City of Hermosillo |

[1] Primary fuel of the largest GPPD match within 5 km

[2] GPPD names have been shortened

[3] Missing in GPPD

Note that a perfect correlation between PP capacity and $NO_x$ emissions cannot be expected, as the emissions per capacity strongly depend on fuel type and technology, and are particularly modified if emission control measures like selective catalytic reduction (SCR) are applied. High emissions for PPs with low capacity probably indicate other dominating $NO_x$ sources nearby dominate, like for Baotou (#6), where a 0.2 GW PP was matching the PS location, but emissions are probably mainly caused by the metal smelting facilities. Low emissions from large PPs probably indicate the installation of SCR or a recent reduction in capacity.

High correlations can be observed for some regions like South Africa and Australia. This is probably indicating that the GPPD entries are reliable for these regions, the level of PP technology is regionally similar, and the divergence method works well here. Fig. 6 (b) displays the scatter plots for coal-fired power plants for Australia, Europe, and South Africa. The slope indicate clear regional differences in the emissions per capacity ratio.


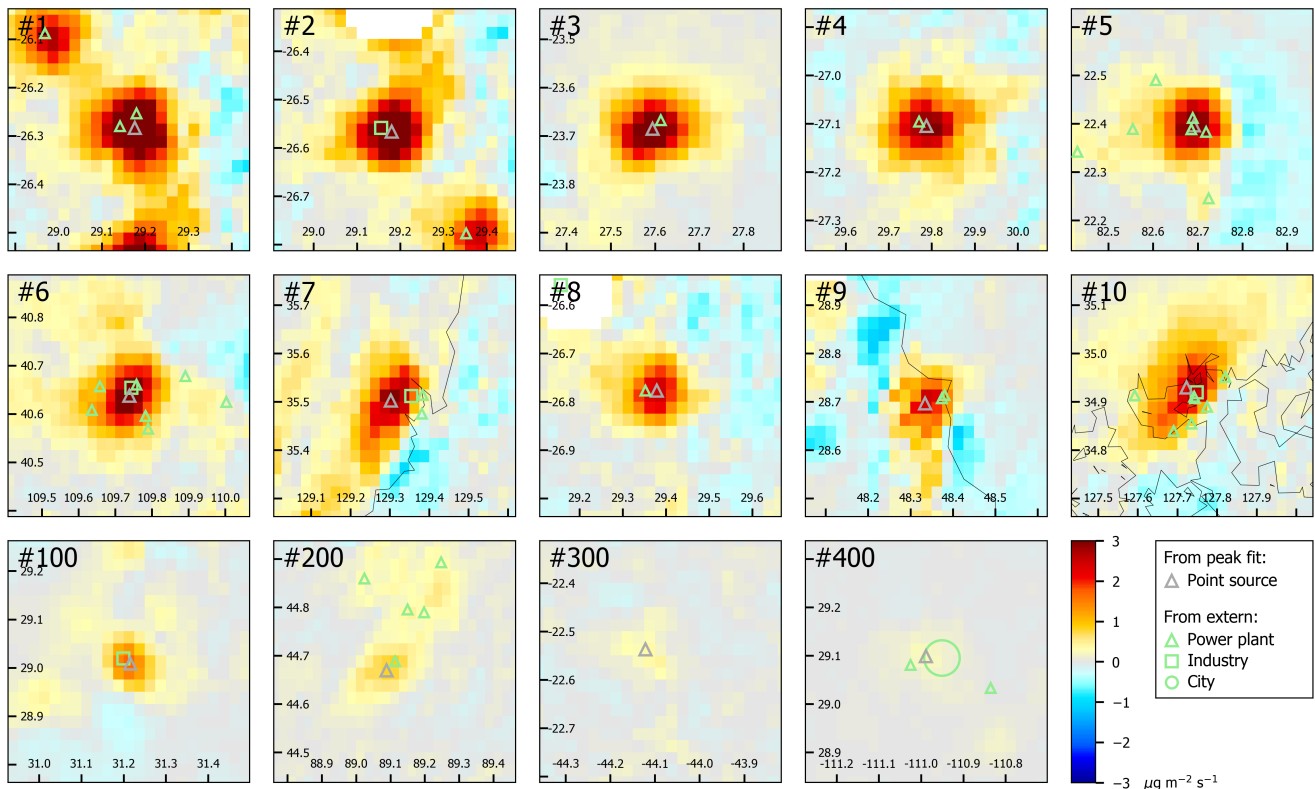

**Figure 5.** Divergence maps for the selected point sources listed in table 4. In addition to the location of the fitted Gaussian peak (grey), also some external information on GPPD PPs as well as industrial facilities and cities is added (green).

## 5 Discussion

In this section we discuss the limitations as well as the potential of the presented point source catalog, give an outline on possible applications, and an outlook on improvements of the catalog in a future update.

### 5.1 Limitations

5   When interpreting the presented point source catalog, the following limitations have to be kept in mind:

#### 5.1.1 Missing point sources

The catalog is incomplete, as point sources might be missing due to the following reasons:

1. Considered pixels:

    Only latitudes between 61° N/S are considered, as for higher latitudes, the strict SZA threshold of 65° would result in

10     poor statistics. In addition, the criteria for defining pixels of interest are quite strict in order to reduce the amount of data



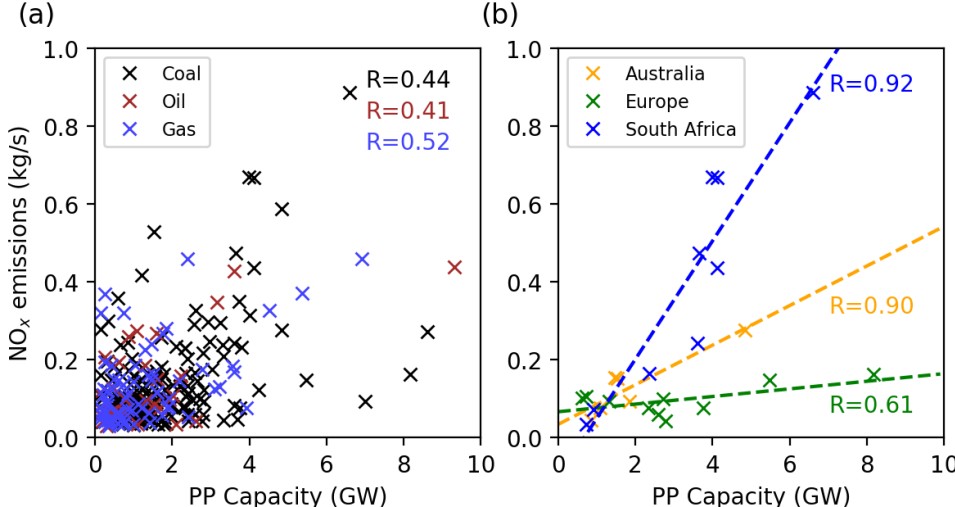

**Figure 6.** Correlation between $NO_x$ emissions and PP capacity. (a) Scatter plot for all matches globally, with color coding primary fuel. (b) Scatter plot for coal-fired power plants in Australia, Europe, and South Africa.

to be processed. There might thus be $NO_x$ point sources not included in the selection mask. However, based on maps of the mean TVCD, we see no indication for strong point sources outside the considered area defined by **M** (compare Fig. 1(a) and (b)).

2. Gaps in input data:

5      The TROPOMI $NO_2$ product (and thus the mean divergence map) reveals persistent gaps at some coastlines, in particular around the Persian Gulf. These gaps are caused by the cloud algorithm, as cloud retrievals are challenging for the transition of dark ocean to bright sand. The situation will improve for an updated cloud product which will be based on a ground albedo with higher spatial resolution.

3. Artifacts in divergence:

10      In case of inaccurate wind fields, as over mountainous terrain, as well as for systematic violation of the steady state assumption, like systematic diurnal cycles of wind direction, the divergence map reveals artifacts, i.e. patterns of high negative values for $D$ which cannot be explained by the loss term $S$. These cases are classified as negative, and are thus missing in the catalog.

4. Noise in divergence:

15      The noise level of the divergence map reveals large regional differences, causing respective differences in the performance of the peak fit algorithm. Noise levels are particularly high over regions with generally high TVCD levels, like Eastern China or Western Europe. This is caused by sampling effects: For high TVCD levels, also fluxes are generally high. As the daily flux maps have gaps due to cloud masking, the mean fluxes reveal "jumps". Note that due to



day-to-day changes of wind directions, a far higher amount of data would be needed in order to get smooth flux maps than to overcome the respective sampling issues in mean TVCDs. As the spatial derivative amplifies these jumps, the divergence is generally noisy over regions with high TVCD. Consequently, only few point sources are identified for the highly polluted regions in Western Europe or Eastern China, while many candidates are classified as negative due to the high noise levels.

5. Interfering sources:

   Point sources cause peaks in the divergence map which can be described by a Gaussian. Typical widths are $\sigma_{x,y} \approx 5\,\mathrm{km}$, in accordance to the spatial resolution of TROPOMI. Thus, point sources within a distance of less than about 20 km cannot be clearly separated by the automated algorithm. In case of point sources close to each other, they are identified and quantified as one single point source (see next section), and their emissions are just added. If the distance is about 15-20 km, however, the joined peak from both sources is still processed as a single candidate, but classified as area source due to the large width of the peak. For example, the Indian power plants Vindhyachal and Anpara located within 16 km (Fig. 3 D) or PP 8 at the western edge of Riyadh Megacity (compare Fig. 2 in Beirle et al., 2019), are classified as area source and thus missing in the catalog.

### 5.1.2 Multiple sources

Due to the spatial resolution of TROPOMI, sources within about 10 km cannot be separated by the automatic peak finding algorithm. Several entries in the catalog contain multiple sources, like #1 (PPs Matla and Kriel within 5 km) or #3 (PPs Matimba and Medupi within 7 km). Similarly, the emissions from large industrial complexes, like the Ulsan industrial area, cannot be further specified. Thus the term "point source" means with respect to the TROPOMI spatial resolution. This is however still better resolved than many bottom-up inventories and typical horizontal resolutions of regional chemical transport models (CTMs). Thus, for emission inventories, also multiple and extended sources could appropriately be treated as PS emissions if used for models with a spatial resolution coarser than that of TROPOMI.

### 5.2 Uncertainties and accuracy

The main goal of v1.0 of the point source catalog is the identification rather than the quantification of $NO_x$ point sources. Still we discuss and try to quantifiy the various sources for uncertainties of the derived point source emissions:

### 5.2.1 Lifetime

The quantification of $NO_x$ emissions is based on the peak in $D$, ignoring the chemical loss of $NO_x$ during downwind transport. We estimate the impact of neglecting the lifetime correction exemplarily for maps of $D$ and $E$ presented in Beirle et al. (2019), and compare the resulting emissions for PPs in Riyadh, South Africa, and Germany. In all cases, the emissions based on $E$ were higher than those based on $D$ by a factor close to 1.10.



The neglection of chemical loss thus causes a slight systematic underestimation of the derived emissions, but the effect of 10% is quite small compared to other effects, in particular the AMF (see below).

### 5.2.2 Wind fields

As discussed in Beirle et al. (2019), different effects on wind field uncertainties have to be considered:

– Errors of the wind direction (both random and systematic) result in a underestimation of the flux, and thus the estimated emissions, as any mismatch in wind direction leads to a low bias of the wind speed component projected to the actual wind direction. The underestimation is thus proportional to the cosine of the wind direction. For wind direction errors of 25°, it is about 10%.

      Larger errors in wind direction would cause visible artifacts in the divergence map and would thus be captured and
10       removed by the check for negative divergence during candidate classification.

   – The calculated fluxes, and thus the divergence map, are proportional to wind speed. The choice of the altitude of input wind fields thus affects the resulting emissions, as wind speeds are generally higher for higher altitudes. In Beirle et al. (2019), the effect for taking wind fields from 250 or 730 m rather than from 450 m on emissions was quantified as about 10%.

In this study, the focus was set to the clear identification of point sources. Thus, winds from a lower altitude (300 m) compared to Beirle et al. (2019) (450 m) were chosen in order to have optimal wind direction data close to the $NO_x$ emission source. As discussed below, this choice removes the artifacts of divergence around the Matimba/Medupi power plants seen in Beirle et al. (2019).

### 5.2.3 Peak fit

The fitted emissions depend on the settings for fit radius and the forward model function. The fit window of 22 km around the source was chosen as compromise in order to cover the peak caused by a point source as far as possible while avoiding interference with neighboring sources. Variations of the fit settings within reasonable limits cause an uncertainty of about 20% in the fitted emissions (Beirle et al., 2019).

### 5.2.4 AMF

Validation studies report on $NO_2$ TVCDs from TROPOMI being biased low (Verhoelst et al., 2020). This is probably due to the a-priori vertical profiles, the cloud height product being biased low, and the surface albedo being biased high, all resulting in high biased AMFs (Eskes et al., 2020).

     In Beirle et al. (2019), a correction of the low bias was applied by assuming the $NO_x$ profile close to point sources being completely in the lowest layer. Note that since the divergence is sensitive for the *change* of the $NO_x$ flux due to the point source
emissions, the required AMF correction has to be applied to the profile of the *added* rather than the total $NO_x$.





Still, we do not apply such a correction here, since there are indications that the cloud heights and surface albedo used for the $NO_2$ retrieval are systematically biased. Thus, also the provided AKs are biased and a simple a-posteriori correction of TVCDs is not possible.

For Germany, the AMF correction applied in Beirle et al. (2019) was a factor of 2 due to profile shape. In combination with a bias in the cloud height and ground albedo used for the TVCD retrieval, even higher factors have to be expected. The actual number will depend on surface albedo, aerosols, and cloud statistics (within the selection of low effective cloud fractions) and is high over dark surfaces and frequent cloud contamination, but low over bright surfaces and few clouds (like for Riyadh). Future updates of the TROPOMI L2 $NO_2$ product will be based on improved albedo and cloud products, and thus more appropriate AKs.

### 5.2.5 Low bias

As listed above, many effects contribute to the uncertainty of emission estimates from the divergence of mean $NO_2$ fluxes. Several of these effects are systematic in nature, resulting in an overall low bias of the derived emissions. In particular the effects of biased cloud height and inaccurate a-priori profiles are expected to reveal large regional dependencies. Consequently, the low bias is hard to quantify for the global catalog. Thus we decided not to try to correct for the low bias of catalog emissions in this study, but present the low biased estimates as they are with a clear disclaimer that the given emission estimates are biased low.

Accordingly, the emission estimates for South Africa reported in Beirle et al. (2019) were higher than the values listed in table 4 by a factor of about 1.87 (for Matla/Kriel) up to 2.56 (for Medupi/Matimba). This discrepancy is a consequence of the missing AMF correction (factor 1.35 applied in Beirle et al. (2019)), the missing lifetime correction (factor 1.1), the wind input data from lower altitudes (factor 1.1), and differences in grid definition, fit function (no rotation), and fit settings (factor 1.2), which together explain a factor of 1.9 as found for Matla/Kriel.

For Medupi/Matimba, the reason for the remaining discrepancy is the difference in input wind fields. The winds from higher altitudes (450 m) used in Beirle et al. (2019) cause a pattern of high negative $D$ southwest of the power plants, indicating a mismatch of wind direction (see Fig. S2B in the Supplement of Beirle et al., 2019). Even in $E$, after applying lifetime correction, the negative values southwest remain in maps of $E$ (Fig. 4A in Beirle et al., 2019). Consequently, the fit function finds a low background with a linear increase from southwest to northeast, and the emissions fitted on top of this low background are biased high.

In the current study, this artifact is not present (see PS #3 in Fig. 5), indicating that the wind direction from lower altitude (300 m) matches better to the actual $NO_x$ transport of Medupi/Matimba emissions.

### 5.3 Potential

Though the catalog is incomplete and the derived emissions are biased low, it still has the potential to improve our knowledge on $NO_x$ emissions. Here we discuss the benefits of the divergence approach and the PS catalog and propose future applications:





### 5.3.1 Localization of $NO_x$ and $CO_2$ sources

The Gaussian fit determines the location of the peak maximum. For all catalog locations which were inspected manually (i.e. all PS listed in table 4 or labeled in Fig. 4), Google Maps quickly reveals a plausible origin of the emissions close to the fitted peak location. For the examples shown in Fig. 5 which are related to PPs, the fitted PS location agrees to the actual PP locations

within about 2-3 km. Thus, the point source catalog accurately lists the location of $NO_x$ point sources. As these sources are all related to combustion, this also provides valuable information on the location of $CO_2$ sources, which may also help to quantify $CO_2$ emissions from current and future satellite measurements of $CO_2$.

### 5.3.2 Spatial patterns

Within a regional focus, the catalog reflects the spatial distribution and relative importance of point sources. Regionally, high

correlation between $NO_x$ emissions and PP capacity was observed (Fig. 6). Thus, it might be possible to re-distribute emissions from bottom-up inventories regionally according to the location of point sources in the catalog.

In addition, striking discrepancies between bottom-up inventories and the catalog, like strong point sources present in one but missing in the other, might be investigated in more detail and should result in improved emission inventories.

### 5.3.3 Up-to-dateness

Bottom-up inventories based on fuel consumption statistics have to collect and process input data from national reports. Thus, they have a time lag and are outdated when released for countries with high dynamics in industrial activities. Based on the divergence of $NO_2$ fluxes, single power plants can be identified and quantified e.g. on annual basis. This would allow to detect short-term trends and PPs being switched on or off even in the vicinity of polluted cities such as Riyadh.

### 5.4 Outlook

This manuscript describes v1.0 of the $NO_x$ point source catalog. We plan to update the catalog as soon as a reprocessed TROPOMI $NO_2$ product is available. We expect that some of the persistent gaps along coastlines, e.g. around Dubai, can be closed in future, as soon as the cloud information is based on high resolution maps of the surface albedo, and thus $NO_2$ TVCD becomes available there. In addition, improved surface albedo, cloud height and a-priori profiles are expected to improve the TVCD and (at least partly) remove the low bias.

In addition, we plan to use meteorological data from ERA-5 on higher spatial and temporal resolution. Currently, 6 hourly model output of ECMWF meteorological data was interpolated to a regular horizontal grid with a resolution of 1°. Case studies will be performed in order to find out which is the best compromise between spatio-temporal sampling and processing time.

## 6 Conclusions

The high spatial resolution provided by TROPOMI allows for the detection and quantification of $NO_x$ "point sources" like power plants, metal smelters, cement plants, or industrial areas. We present v1.0 of a global catalog of $NO_x$ point sources derived from the divergence of the mean $NO_2$ flux 2018-2019 by a fully automated iterative algorithm, yielding 451 point sources. 242 of these point sources could be matched to combustion power plants from a global database, with a median distance of 1.6 km. Top 4 point source emissions are all located in South Africa and related to coal burning. About 1/4 of all point sources were found over the Indian subcontinent.

The catalog misses some point sources due to gaps in the divergence map (caused by gaps in the cloud product), artifacts in the divergence map (caused by non-steady state and inaccurate wind fields), noise in the divergence map (caused by sampling effects for regions with high background TVCD), and interference of sources within about 10-20 km distance.

The listed emissions are biased low mainly due to a low bias of input TVCDs from TROPOMI. As this bias is expected to vary regionally and is hard to quantify, it is not corrected for v1.0 of the catalog.

Still, the catalog has high potential for checking and improving emission inventories, as it localizes $NO_x$ (and also $CO_2$) sources, provides spatial patterns of the distribution of sources, and yields up-to-date emission data.

## 7 Data availability

The full $NO_x$ point source catalog v1.0 is available at https://doi.org/10.26050/WDCC/Quant_NOx_TROPOMI (Beirle et al., 2020). The corresponding divergence maps are provided by the authors on request.

*Author contributions.* SB performed the analysis and wrote the manuscript with input from all co-authors. CB processed the TROPOMI L2 data. SD processed ECMWF meteorological data. HE performed the retrieval of operational TROPOMI L2 $NO_2$ TVCDs. VK compiled the Ozone climatology from ESCiMo model data. JdL initiated the matching between catalog point sources and GPPD. TW contributed to the data analysis and interpretation and supervised the study.

*Competing interests.* None

*Acknowledgements.* We thank ESA and the TROPOMI L1/L2 teams for realizing TROPOMI and providing $NO_2$ tropospheric data. ERA Interim and ERA5 data used in this study are provided by the European Center of Medium-Range Weather Forecasts (ECMWF). Andrea Pozzer is acknowledged for providing the ESCiMo model data used for the Ozone climatology.





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
