# Peer review of "Catalog of $NO_x$ emissions from point sources as derived from the divergence of the $NO_2$ flux for TROPOMI"

_Earth System Science Data, 2020_

## Author Comment (AC1)

We would like to thank reviewer #1 for the positive assessment, constructive criticism and helpful comments. Below we reply to the raised issues point by point, where the comments from the reviewer are typed italic.

*This is a nice manuscript that describes an initial attempt to estimate NOx emissions from 100s of point sources around the globe using TROPOMI NO2. It is well-written and includes a lot the detailed information necessary for replication. Some minor comments are below. The three more major comments have an asterisk in front of them.*

*Page 3 Line 5. Please note here that section mask will be described in more detail in Section 3.3*

The explanation of the section mask caused some confusion with the actual data processing for the divergence calculation. In order to avoid this confusion, we moved the complete section about the selection mask to the Supplement (see below).
In the manuscript, we have modified the respective sentence into
"Thus, the analysis is only performed around stationary NOx sources, which are defined based on the magnitude as well as the temporal variability of TROPOMI TVCDs."
We would prefer not to add a concrete a detailed reference to a particular section in this rather general paragraph of the introduction.

*Page 5 Line 26. Use a specific value, perhaps "less than 5 km" instead of "some km"*

We modified this sentence to
"In order to have maximum sensitivity for point sources, the TROPOMI observations have to be oversampled, requiring a fine grid resolution of less than 3 km." and shifted it to section 3.2 (grid), since the selection mask is now described in the Supplement.

*Figure 1a. Include a "+" at the end of the colorbar, as I am assuming there is a value somewhere >10x10^15*

We have modified the colorbars for all figures such that it becomes clear that values could also be higher than the indicated range.

*\*Page 6 Line 1. It's still a bit unclear how the TROPOMI data is averaged and filtered. Are entire months excluded if there are only 4 valid overpasses instead of 5? Or is this just a selection criteria. Also are the emissions derived monthly (or some group of months together) and then averaged over the 24-month period? Or is the TROPOMI data averaged over all 24 months and then the emissions are computed? I'd imagine the former might be better for mid-latitudes since NOx lifetime varies quite a bit due to seasons.*

We are sorry that our description was not clear enough. In the revised manuscript, we tried to clarify the raised issues.
In fact, we processed the complete TROPOMI time series twice:
- In the first processing, the only aim is to find potentially stationary sources and define the mask. Here, all valid TROPOMI measurements up to SZA 80° are included. Only for this first processing, monthly means (and maxima and minima) are evaluated.
- In the second processing, only valid TROPOMI pixels up to SZA 65° within the defined mask are gridded. Only for this selection, the further steps (upscaling to NOx, calculation of fluxes) are performed.
In order to avoid confusion, we decided to move the description of the selection mask to the Supplement.

*Page 6 Line 10. Is it 6 months during each year (12 month period) or 6 months during the full 24 months?*

We clarified this into:
"… for at least 6 months during the 24 month period 2018-2019."
This section is shifted to the Supplement now.

*Figure 2 caption. Prefer if "and" is replaced by "divided by". Presumably values shown here are a value averaged over all 24 months? Or only "valid months" shown? Also, would be good to state either in the figure caption or text accompanying the figure that polar regions have more interannual variability in the NOx lifetime at the TROPOMI overpass time.*
*Lastly, while larger ratios likely exist over mountainous regions, I am not convinced that this particular image is showing that. Instead it seems like latitude is the primary driver of the variability. I suggest removal of the last sentence, and replacement with a comment such as "latitude (and associated confounders) is one of the primary drivers of the NOx/NO2 ratio".*

The shown values are the ratio of temporal mean NOx divided by temporal mean NO2 for the final selection of TROPOMI pixels, which should be clear in the revised manuscript, as confusion is avoided by shifting the description of the selection mask to the supplement.
We agree that the statement on mountains is not fully supported by Fig. 2, as e.g. values over the Rocky Mountains are not increased. Thus, we skipped this statement.
For an annual mean including all seasons, one would indeed expect that the NOx/NO2 ratio depends strongly on latitude. Here, however, the selection of SZA<65° removes wintertime measurements for mid/high latitudes. This is written in the text, but is now also pointed out in the figure caption of the revised manuscript.
There is a remaining tendency of higher ratios for higher latitudes, but the zonal variability can be similarly high. Thus we refrain from pointing out the latitude as primary driver, but clarify why the latitudinal dependence is not as strong as expected.
In order to clarify these aspects, we have modified the figure caption to
"Effective NOx/NO2 ratio, i.e. mean tropospheric NOx (as derived by assuming photostationary state according to Eq. 2 for each TROPOMI pixel) divided by the mean NO2 column density for 2018-2019. Note that only cloud free observations with SZA<65° are considered, thus wintertime measurements at mid to high latitudes are skipped, and the expected latitudinal dependency of the NOx/NO2 ratio is suppressed. The spatial variability is a consequence of the dependency of the photostationary state on actinic flux, Ozone concentration, and temperature (Eq. 2). "

*Page 10 Line 5. Emphasize that latitude appears to be one of the largest controllers of NOx lifetime at the TROPOMI overpass time. Can you further comment on how this may bias your results for different areas? If I understand correctly, a region with a shorter NOx lifetime, might have a larger sink value, which is excluded by your method. So perhaps excluding the sink term may cause a larger bias in equatorial regions than mid-latitudes. Even if my understanding is incorrect please comment on how latitude affects NOx lifetime and any associated biases.*

Note that also the latitude dependency of the mean lifetime is suppressed by the selection of SZA < 65°, i.e. the implicit skipping of wintertime at midlatitudes. For all considered (cloud free) TROPOMI observations, solar irradiance is relatively high. We point this out in the revised manuscript.
Concerning the impact of short lifetimes: A very short lifetime would indeed result in high negative divergence, in particular for point sources with stable wind direction, as negative values accumulate in downwind direction. This might cause the unintentional skipping of some point sources by classifying them as "negative".
For the calculation of emissions, however, negative values of *D* are not excluded, but are considered in so far that the fit allows for a constant background. I.e. a power plant downwind from an area

source (when divergence would be raised quite homogeneously by the lifetime correction) should not be affected strongly.

But indeed, the derived emissions might be affected if the lifetime is very short. In the discussion manuscript, we only estimated the effect of omitting the lifetime correction for an a-priori value of 4h for tau, which results in a small effect (10%) for the examples we have checked in the initial manuscript. We have now checked the impact of the missing lifetime correction for all point sources, and found that the effect is indeed larger on average (25%).

And if the lifetime would be as short as 1.5 h, as reported by Goldberg et al. (2019) for the Colstrip power plant (Fig. 1 therein), the emission estimates based on lifetime corrected $D$ would be even higher by a factor of 1.73.

We have included this aspect into the discussion of the missing lifetime correction (5.2.1) and updated the uncertainty estimates.

*Page 10 Line 27. Would be good to give reader an estimate of the low bias, even if it is a range. Seems like bias can be anywhere from a factor or 1.2 to almost 2.5. Goldberg et al., 2019 discusses this a bit for U.S. power plants using the exponentially modified Gaussian fit (Beirle et al., 2011). Goldberg, D. L., Lu, Z., Streets, D. G., De Foy, B., Griffin, D., Mclinden, C. A., Lamsal, L. N., Krotkov, N. A. and Eskes, H.: Enhanced Capabilities of TROPOMI NO2 Estimating NOx from North American Cities and Power Plants, Environ. Sci. Technol., 53(21), doi:10.1021/acs.est.9b04488, 2019.*

The low bias is discussed in detail in section 5.2.5. We have extended this discussion significantly and now also provide some more concrete numbers. In addition, we also give a rough number already in section 3.7 that are reported in the cited refences.

We included a reference to Goldberg et al., 2019 in the discussion of lifetime effects. For the low bias, however, Goldberg et al. is probably not the ideal reference, as they corrected for the low bias by re-calculating AMFs based on GEM-MACH model profiles.

*Page 11 Line 20. What is meant by the "Global maximum value"? Please clarify.*

We modified this phrase to "the initial maximum value of the first iteration".

*Figure 3 caption. Over what timescales are being shown here?*

We modified the caption to "Zooms of the mean divergence (2018-2019) …"

*Page 17 Line 12. Any chance the csv file can be included as a supplement on the ESSD webpage or another publicly available webpage? Seems like you need an approved account to access the csv file, which might hinder some people from accessing it.*

We will try to add the csv file as supplement to the manuscript directly, if permitted by ESSD.

*\*Page 17 Line 15. Instead of or addition to every "100th", perhaps include the largest on each continent and their rank. Table 4 and Figure 5 have no values for US, Europe or Australia (and North America and South America only appear because they have a "100th"). As of right now, it's difficult to assess how well the method does in North America, South America, Europe, and Australia (these 4 continents likely represent a large segment of the journal readership).*

We thank the reviewer for bringing this up, and we fully agree that results for North America and other regions are underrepresented in the selected figures and tables. Thus we have added regional tables of the catalog (like table 4) of the strongest point sources for all considered regions to the supplement.

*Page 18 Line 5. Modify "large" to "high capacity"*

Done.

*Page 21 Line 21. Explicitly state that in your method, the reported emissions are a grouping of emissions from electricity generation and any related emissions from on-site activities such as heavy-duty diesel, etc. The latter may be small, but important to make this distinction.*

We have added the statement "Power plant stack emissions would automatically be merged with emissions from on-site activities such as heavy-duty diesel around the power plant."

*\*Page 21 Line 25. I understand that accurate quantification isn't necessarily the goal here, but it's still important to compare to reported NOx emission data. In the United States, at least, annual NOx emissions data from top power plants is very easy to acquire. To download annual data visit this website: https://ampd.epa.gov/ampd/ , scroll down to "Get Reports" on bottom right and click "Start", on next page click "Top Emitters", then choose Annual NOx emissions for 2018 (and 2019) by Facility for the top 50 emitters. Seems like European data is here: https://www.eea.europa.eu/data-and-maps/data/industrial-reporting-under-the-industrial-2 I recommend including a discussion in this section comparing your reported values to the values reported by the US and European agencies.*

Following the reviewer's proposal, we have checked the 10 top emitters for the USA in 2019. 7 out of these 10 power plants are listed in the point source catalog, with correct naming derived from the merging of GPPD information. For the 3 missing plants, also a peak in the divergence map is visible, but the automated algorithm classified these all as "negative", i.e. a large negative divergence is observed around the plants. As explained in section 3.8.1, this might indicate high noise levels of $D$ or systematic artefacts which might be caused by biased wind fields. Thus, these cases are excluded within the automatized point source identification.

[Figure]

Fig. R1: Zooms of the divergence map around (+/- 50 km) the top ten emitters of the USA. This figure is added to the revised Supplement.

When comparing the emissions, however, we find quite large discrepancies:

Table R1: Top ten NOx emitters for the USA in 2019 according to EPA and the corresponding catalog emissions. This table is added to the revised Supplement.

| Name | Emissions (EPA) (kg/s) | Emissions (this study) (kg/s) |
|---|---|---|
| New Madrid | 0.446 | 0.074 |
| Colstrip | 0.432 | 0.079 |
| Miami Fort | 0.360 | 0.053 |
| Navajo | 0.351 | 0.115 |
| Hunter | 0.333 | 0.040 |
| Scherer | 0.319 | - |
| Martin Lake | 0.301 | - |
| Fort Martin | 0.298 | - |
| Intermountain | 0.287 | 0.054 |
| Thomas Hill | 0.285 | 0.037 |

For Navajo, which yields the most obvious signal in the divergence map, and is #153 emitter of the global catalog (#3 of North America), the EPA emissions are higher by a factor of 3, which is well in the range of the expected low bias mainly due to missing AMF and lifetime correction.

For the other power plants, however, EPA value are higher by factors of 5-8, which is out of the range of uncertainty given in the discussion paper. These power plants, however, are quite remote from large cities. Thus, in absence of other significant NOx sources, the modelled profiles used as a-priori for the calculation of AMFs do not reflect the near-surface power plant plume due to the coarse model resolution.

In addition, the cloud altitude used for the calculation of averaging kernels is biased low (Compernolle et al., 2021), causing high biased AMFs and low biased columns. The impact of this bias can easily reach one order of magnitude for cases where the retrieval assumes a cloud below the plume, while it is actually above.

For these reasons, we have to expect that the low bias of the partial column added by a point source close to the ground is considerably larger than that of the complete tropospheric column, where validation studies report typical low biases of a factor of 2 for polluted sites.

We will focus on this issue more quantitatively when preparing an update of the catalog, which we plan to process after reprocessing of the TROPOMI NO2 product with improved cloud products and thus more appropriate AKs.

We have added the comparison to the top 10 USA emitters shown above to the Supplement.

In the revised manuscript, we made the following additions/modifications:

Abstract:
"The derived emissions are generally too low, lacking a factor of about 2 up to 8 for extreme cases. This strong low bias results from combination of different effects, most of all a strong underestimation of near-surface NO2 in TROPOMI NO2 columns."

Section 5.2.5:
"For partly clouded scenes, the AMF for the added NO2 column could be easily too high by an order of magnitudes, if the real cloud is above the plume, but the retrieved cloud is below. Consequently, the added NO2 would be biased low by an order of magnitude."

Section 5.2.6:
"In the Supplement, catalog emissions are exemplarily compared to emissions reported by EPA for the top 10 emitters of the USA. 7 of these 10 emitters are listed in the catalog, with correct naming

found from the merging of GPPD. The other 3 emitters were also found as candidates, but were classified as 'negative'.

EPA emissions were found to be higher than the emissions listed in the catalog by a factor of 3 (Navajo) up to 8 (Hunter). These power plants, however, are quite remote from large cities. Thus, in absence of other significant NOx sources, the modelled profiles used as a-priori for the calculation of AMFs do not reflect the near-surface power plant plume due to the coarse model resolution.

In addition, the cloud altitude used for the calculation of averaging kernels is biased low (Compernolle et al., 2021), causing high biased AMFs and low biased columns. The impact of this bias can easily reach one order of magnitude for cases where the retrieval assumes a cloud below the plume, while it is actually above.

For these reasons, we have to expect that the low bias of the partial column added by a point source close to the ground is considerably larger than that of the complete tropospheric column, where validation studies report typical low biases of a factor of 2 for polluted sites.

We will focus on this issue more quantitatively when preparing an update of the catalog, which we plan to process after reprocessing of the TROPOMI NO2 product with improved cloud products and thus more appropriate AKs."

Conclusions:
"Exemplary comparisons to emissions reported from in-situ monitoring reveals a low bias which can be as high as a factor of 8 for some power plants, which is probably caused by inappropriate a-priori profiles and a low bias in the cloud height."

*Page 24 Line 7. Would be good to cite Liu et al., 2020 here. Liu, F., Duncan, B. N., Krotkov, N. A., Lamsal, L. N., Beirle, S., Griffin, D., McLinden, C. A., Goldberg, D. L. and Lu, Z.: A methodology to constrain carbon dioxide emissions from coal-fired power plants using satellite observations of co-emitted nitrogen dioxide, Atmos. Chem. Phys., 20(1), 99–116, doi:10.5194/acp-20-99-2020, 2020.*

Thanks for the hint. We have added a reference to this manuscript here.

*Page 25 Line 7. Important to note in this section that there are plenty of point sources in North America and Europe, but many are generally too small to capture in your current method (i.e., blend in with background NO2 or other local sources).*

We fully agree that this aspect is an important caveat of the catalog, which is already stated in the paragraph that follows. We modified this section in order to make this more clear.

---

## Author Comment (AC2)

Mainz, 13. May 2021

We would like to thank reviewer #2 for the positive assessment, constructive criticism and helpful comments. Below we reply to the raised issues point by point, where the comments from the reviewer are typed italic.

*General comments:*

*This manuscript describes a unique data set of anthropogenic NOx point emissions derived from nearly two years of TROPOMI data, explains the methods used to create it, and discusses its limitations and uncertainties.*

*The data set presented is limited in several ways: it is incomplete because of the method used, it is only partly quantitative as it suffers from a known but poorly constrained low bias in the current TROPOMI lv2 data, and it is potentially biased because the assumption has to be made, that emissions are constant over the time period used as input. Therefore, this data set should not be used as emission input for CTM runs or for the verification of reported emissions.*

*In spite of these limitations, this is a very valuable data set as it provides observational evidence for the position and approximate strength of a large number of NOx point emissions sources worldwide. It is a unique data set and fully independent of other data sets of NOx point sources and therefore very well suited for semi-quantitative verification of other inventories, in particular if up-to-date information is needed.*

*The data set has the potential to be improved by future updates when more TROPOMI data is available for better statistics, and when improved TROPOMI lv2 data versions have been released for reduction of uncertainties.*

*The manuscript describes the generation of the data set in a clear and sufficiently detailed way, highlighting the differences to the scientific study describing the original method. The uncertainties and limitations are discussed in detail and give users a good idea of how to use the data, and which caveats to consider.*

*Detailed comments:*

*The authors introduce a number of ad hoc abbreviations such as PP for power plants, which in my opinion make the text less accessible and do not really help with length. I would suggest using them only in the figures.*

We now write out "power plants" and "point source" throughout the document and skip the respective abbreviations.

*Page 2, line 7 – 9: I am confused by this description of AMF and Averaging Kernel. To me, this should be Jacobians or box AMFs instead of Averaging Kernels as the latter are the box AMFs divided by the total AMF derived for the a priori profile. Please reconsider.*

For the introduction, we simplified this sentence into
".. and applying the so-called air mass factor (AMF) that depends on the NO2 profile shape as well as on viewing geometry, surface albedo, aerosols, and particularly on clouds."
In addition, we extended section 3.7 ("AMF correction") by introducing box-AMFs.

*Page 2, line 31: "flux increases discontinuously" – I find this formulation a bit awkward as a discrete data set is discontinuous by definition*

We modified this phrase to "the NOx flux increases abruptly".

*Page 3, line 18: TROPOMI spatial resolution has already been introduced*

This is indeed some redundancy. Still we think that it is important to give the resolution already in the introduction, but would like to keep it in the section about TROPOMI as well.

*Page 3, line 27: Mention native resolution of model data*

We revised this paragraph to
"Until August 2019, ERA-Interim data are used with a truncation at T255, corresponding to ~0.7° resolution. Since September 2019, ERA-5 data are used with a truncation at T639, corresponding to ~0.3° resolution."

*Page 4: Mention altitude of O3 extraction*

As the divergence is sensitive for the added NOx at the source, the relevant NOx/NO2 ratio is that close to ground. We thus took O3 concentrations from the lowest model layer and added this information to the revised manuscript.

*Page 5: More details on the gridding would be useful: I assume it is a 2d linear interpolation?*

Yes. We added this information to the manuscript.

*Is there a risk of low bias using this approach? Is this approach reducing the spatial resolution of the original data? Mention that is done by orbit (this is explained later but I asked myself here if this is done daily, monthly or on orbits)*

We thank the reviewer for raising these issues. We have checked exemplarily how far the gridding by linear interpolation might introduce a low bias. For the power plant plume of PP9 northeast of Riyadh on 17 December 2017 (Fig. 1a in Beirle et al., 2019), the average plume VCD from interpolation yields almost the same value than for conventional gridding (1% lower). For the peak maximum, however, which is more relevant for the divergence than the mean, interpolation is 6% lower. We have added this aspect to section 5.2 (Uncertainties and accuracy).
As we use a grid with high resolution, we see no loss of the original data resolution.
In the revised manuscript, we clearly state that all processing steps are done by orbit, and hope that confusion with monthly means is now avoided by shifting the description of the selection mask to the Supplement (see also reply to reviewer #1).
One further aspect related to the 2D-interpolation, which we noticed during the case study above, is the occurrence of gaps, e.g. due to clouds. Missing values have to be set to NaN, as already stated in section 3.2. This results in gaps in the gridded product that are far larger than for conventional gridding. This is probably also one reason for the poor statistics over e.g. Germany. We added this aspect to section 3.2.

*Page 7: Is there a risk that the requirement for a high dynamic range excludes some point sources with little variation in wind direction?*

For a point source with persistent wind, this indeed might principally happen. However, already slight variations in wind speed or direction are sufficient to cause high variability of the column density around the source at least for some grid pixels.

From the resulting selection mask, which includes all potentially stationary sources plus 1° around, we don't see indications that significant NOx sources are missing.
Still, we might choose a less strict selection mask for a future update of the catalog.

*Page 9: The upscaling to NOx would benefit from a bit more discussion – NO2 columns are inserted where NO2 concentrations should be used, and in particular when [O3] is taken at varying altitudes, this raises some questions about the validity of the approach. For high altitude regions, the factor between NO2 column and NO2 concentration will also change – does this have an effect?*

We have modified the respective paragraph as follows, with additions in bold type:
"**Near-surface** Ozone mixing ratios are taken from a climatology based on the ESCiMo model simulation and converted into concentrations based on T and p from ECMWF.
The derived values for L represent conditions for surface-near pollution. For background NOx in the upper troposphere, the partitioning would be shifted towards NO.
However, any additive background is automatically removed by the calculation of the divergence.
**Thus, the partitioning derived for near-surface concentrations is appropriate also for correcting the added column caused by a point source.** "

*Page 10, line 22: Is this really the best reference here – Eskes et al. introduced the column AK, not the AMF.*

We revised this paragraph and added the term "box AMFs", plus a reference to Wagner et al., 2007:
Wagner, T., Burrows, J. P., Deutschmann, T., Dix, B., von Friedeburg, C., Frieß, U., Hendrick, F., Heue, K.-P., Irie, H., Iwabuchi, H., Kanaya, Y., Keller, J., McLinden, C. A., Oetjen, H., Palazzi, E., Petritoli, A., Platt, U., Postylyakov, O., Pukite, J., Richter, A., van Roozendael, M., Rozanov, A., Rozanov, V., Sinreich, R., Sanghavi, S., and Wittrock, F.: Comparison of box-air-mass-factors and radiances for Multiple-Axis Differential Optical Absorption Spectroscopy (MAX-DOAS) geometries calculated from different UV/visible radiative transfer models, Atmos. Chem. Phys., 7, 1809–1833, https://doi.org/10.5194/acp-7-1809-2007, 2007.

*Page 11: I would suggest to already here identify x and y as latitude and longitude*

x and y have been defined as distance along longitude/latitude in section 3.2.

*Page 12, line 15: neglects of the => neglects the*

Corrected.

*Page 14, line 27: above > 30% => above 30%*

Corrected.

*Page 15: Maybe in India, there is a lack of distributed NOx emissions such as from traffic, helping to reduce the background levels*

From Fig. 1 it can be seen that background levels of NO2 in India are lower than in China or Western Europe, but they are not really low. Thus we are reluctant to write about a "lack of distributed NOx emissions" over India.
One important difference between India and e.g. Germany is the strong seasonality. In India, the dry season provides very good observation conditions, while for Germany, the frequent occurrence of patchy clouds regularly cause gaps in the input data which are magnified by the 2D interpolation.
We have added this aspect to the revised manuscript.

*Page 17, line 7: Ukraina => Ukraine*

Corrected.

*Page 22, line 9: I am not convinced that larger deviations of wind direction will be readily visible in the divergence as this is applied to the averaged flux where wind direction errors could cancel*

We agree that errors would at least partly cancel for statistical deviations. We modified the sentence to
"Larger systematic errors in wind direction would cause visible artifacts in the divergence map and would thus be captured and removed by the check for negative divergence during candidate classification. In case of larger random effects, the artefacts of individual days would at least partly cancel in the mean flux. But the wind speed component in the actual wind direction would be significantly underestimated, as well as the resulting divergence."

*Page 25, acknowledgements – isn't there a standard statement required when using EU Copernicus data?*

We have used TROPOMI data provided by ESA and wind fields from ECMWF, as already mentioned in the acknowledgements. We have extended the acknowledgements and also added the doi for TROPOMI NO2 data. However, we are not aware of a standard text required for the datasets we are using. If our acknowledgements are still insufficient, we would appreciate a concrete hint about what is missing.